# Clade-D auxin response factors regulate auxin signaling and development in the moss *Physcomitrium patens*

**Carlisle Bascom, Jr.**[iD]**, Michael J. Prigge, Whitnie Szutu, Alexis Bantle, Sophie Irmak, Daniella Tu, Mark Estelle**[iD]*****

Department of Cell and Developmental Biology, School of Biological Sciences, University of California San Diego, San Diego, California, United States of America

* mestelle@ucsd.edu

## Abstract

Auxin response factors (ARFs) are a family of transcription factors that are responsible for regulating gene expression in response to changes in auxin level. The analysis of ARF sequence and activity indicates that there are 2 major groups: activators and repressors. One clade of ARFs, clade-D, is sister to clade-A activating ARFs, but are unique in that they lack a DNA-binding domain. Clade-D ARFs are present in lycophytes and bryophytes but absent in other plant lineages. The transcriptional activity of clade-D ARFs, as well as how they regulate gene expression, is not well understood. Here, we report that clade-D ARFs are transcriptional activators in the model bryophyte *Physcomitrium patens* and have a major role in the development of this species. $\Delta arfd^{dub}$ protonemata exhibit a delay in filament branching, as well as a delay in the chloronema to caulonema transition. Additionally, leafy gametophore development in $\Delta arfd^{dub}$ lines lags behind wild type. We present evidence that ARFd1 interacts with activating ARFs via their PB1 domains, but not with repressing ARFs. Based on these results, we propose a model in which clade-D ARFs enhance gene expression by interacting with DNA bound clade-A ARFs. Further, we show that ARFd1 must form oligomers for full activity.

## Introduction

The phytohormone auxin is a key regulator of many developmental processes in plants. Auxin is perceived within the nucleus by a co-receptor complex consisting of a TRANSPORT INHIBITOR RESISTANT 1/AUXIN F-BOX (TIR1/AFB) protein, the substrate binding subunit of an SCF E3 ubiquitin ligase complex, and an AUXIN/INDOLE-3-ACETIC ACID (Aux/IAA) protein, a transcriptional repressor. Auxin promotes the TIR1/AFB and Aux/IAA interaction resulting in Aux/IAA ubiquitination and degradation [1]. Once the Aux/IAA proteins are degraded, transcription factors called AUXIN RESPONSE FACTORS (ARFs) activate transcription of auxin-responsive genes [2,3].

ARFs are classed as transcriptional activators or repressors based on experimental evidence as well as sequence homology [3,4]. Most ARFs have 3 domains. Near the N-terminus is a B3

**Funding:** This work was supported by grants from the National Institute of General Medical Sciences (R01 GM043644 to ME; R35GM141892 to ME). CB and MP received salary from GM141892. CB, MP, WS, SI, and DT received salary from GM043644. The funders had no role in study design, data collection and analysis, decision to publish, or preparation of the manuscript.

**Competing interests:** The authors have declared that no competing interests exist.

**Abbreviations:** ARF, auxin response factor; BRD, B3 repression domain; DBD, DNA-binding domain; IAA, indole acetic acid; MR, middle region; NAA, 1-naphthaleneacetic acid; YFP, yellow fluorescent protein.

DNA-binding domain (DBD) [2]. Adjacent to the DBD lies a large middle region (MR). In the case of clade-A ARFs, the MR is enriched in intrinsically disordered sequences and is required for transcriptional activation [3]. Finally, most ARFs have a C-terminal Phox/Bem1 (PB1) domain that facilitates interaction with Aux/IAA repressors as well as other ARFs [1]. PB1 domains consist of separate positively and negatively charged surfaces (formerly DIII and DIV, respectively) that facilitate oligomerization through surface charge interactions in a head to tail fashion [5,6].

Several recent studies have focused on the conserved elements of auxin signaling in bryophytes [7–9]. Much of the auxin signaling molecular architecture is conserved in the model moss *Physcomitrium patens* [10,11]. Auxin is involved in several developmental processes including reproductive organ formation [12], phyllid expansion [13], and colonization by tip-growing cells [14]. Haploid *P. patens* spores germinate to form filaments comprised of cells called protonemata. Protonemal filaments lengthen via divisions of a tip-growing apical cell. Early in development, the apical cell is slow growing and chloroplast-rich, generating short cells with perpendicular cross-walls as it divides. These cells are referred to as chloronemata. Within days, the apical cell transitions to a fast-growing, chloroplast-poor caulonemal cell. These cells are longer than chloronemal cells, and their cross-walls are at an oblique angle to the axis of growth. The chloronema–caulonema transition is promoted by the auxin indole acetic acid (IAA) based on work in suspension cell culture with another moss, *Funaria hygrometrica* [15]. Asymmetrical cell divisions of caulonemal cells give rise to gametophores. Under specific environmental cues, gametophores develop gamete-producing gametangia to complete the life cycle [16,17].

Studies of the auxin co-receptor AFB proteins as well as the transcriptional repressor Aux/IAAs provide further evidence for auxin's role in *P. patens* development. RNAi knock-downs of the *PpAFB* genes results in plants with striking growth defects including the lack of recognizable caulonemata [11]. Similarly, gain-of-function mutations that stabilize Aux/IAAs result in plants with repressed auxin signaling. Developmentally, plants expressing stabilized Aux/IAAs exhibit a delay in the chloronemata-to-caulonemata transition, resulting in an inability to make gametophores [11]. Interestingly, *P. patens* plants lacking *Aux/IAA* transcriptional repressors, and thus constitutively responding to auxin, make protonemata that could not be clearly identified as either chloronemata or caulonemata [18]. This suggests that repression of some auxin-response genes is required to generate both chloronemata and caulonemata. In total, these studies highlight the importance of auxin to the chloronema–caulonema transition and other aspects of moss development.

The role of individual ARF proteins in bryophyte growth and development is an active area of study. The liverwort *Marchantia polymorpha* has a single activating ARF (MpARF1), a single repressing ARF (MpARF2), and one clade-C ARF (MpARF3) that acts as an auxin-independent transcription factor [9,19]. MpARF1, 2, and 3 each have a single DBD, MR, and PB1 domain. In addition to these proteins, *M. polymorpha* encodes a noncanonical ARF called *MpncARF* that lacks a DBD [20]. Despite the absence of a DBD, ncARF is involved in auxin-mediated gene regulation. Several auxin-sensitive genes are misregulated in the mutant, indicating that ncARFs play a role in the auxin-signaling pathway [20]. However, loss of ncARF did not result in an obvious developmental defect.

In contrast to *M. polymorpha*, the *P. patens* genome encodes 7 clade-A, 4 clade-B, 3 clade-C, and 2 clade-D ARFs [18,21]. The *P. patens* clade-D ARFs are homologous to *MpncARF* and also lack DBDs. Since clade D nomenclature is consistent with the widely accepted clade-A, -B, and -C designation, we propose the use of clade-D (ARFd) to describe this group of ARF proteins [18–20]. Phylogenetically, D ARFs are in a sister clade to activating A ARFs [18–20], suggesting that they may function as activators. However, their mode of action and their role in *P. patens* development is unknown.

Here, we show that the PpARFds play a significant role in auxin response and development. Further, we present evidence that multimerization of the ARFds is key to their function. Lastly, we show that auxin signaling in protonemata plays a role in filament branching, and that the PpARFd's are required for this process.

## Results

### A screen for auxin-resistance identifies mutations in ARFd1

To identify novel components of the *P. patens* auxin-signaling pathway, we screened for mutants resistant to the synthetic auxin, 1-naphthaleneacetic acid (NAA). Seven-day-old tissue regenerated from a homogenized Gransden ecotype strain of *P. patens* containing the *DR5: DsRed2* auxin-responsive reporter as well as *35S:NLS-GFP-GUS* [22] gene (DR5:*DsRed2 Gd/NLS4)* [18] was mutagenized using UV light. This tissue was then transferred to medium supplemented with NAA and visually screened for NAA-resistant (*nar*) phenotypes. Wild-type plants produce rhizoid-like cells on this medium, whereas *nar* mutants produce green protonemal tissue or leafy gametophores (Fig 1A). Although we planned to use the *DR5:DsRed2* reporter as an additional readout for auxin response, we found this reporter to be unreliable.

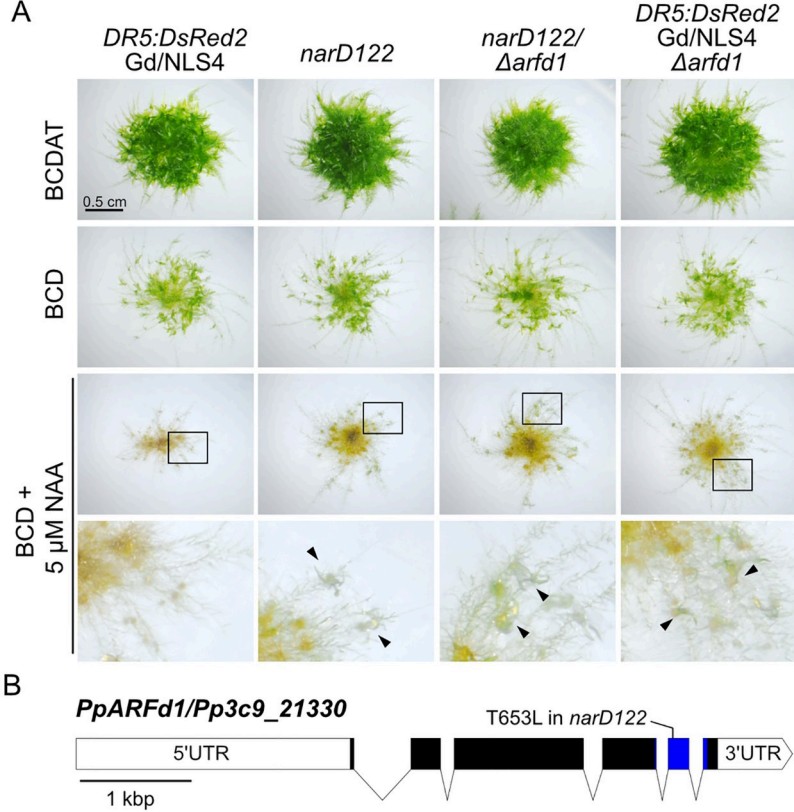

**Fig 1. Isolation of *NAA-Resistant* (*nar*) mutants.** (A) Representative micrographs of 21-day-old spotted colonies on indicated medium. *narD122* was isolated from irradiated *DR5:DsRed2*/GdNLS4 tissue. *narD122* has a T653L mutation in *ARFd1*. Knocking out *ARFd1* in *narD122* background does not restore auxin sensitivity, suggesting that *arfd1*^T653L^ is not a dominant mutation. Knock out of *ARFd1* in the *DR5:DsRed2/GdNLS4* line reduces auxin sensitivity. Both *narD122* and *Δarfd1* lines produce leafy phyllids on BCD + 5 μM NAA (black arrowheads). (B) The *PpARFd1* locus. UTR is the untranslated region. Blue region corresponds to PB1 domain. ARF, auxin response factor; NAA, 1-naphthaleneacetic acid.

Fifty-seven mutants were isolated including 18 new alleles of previously identified *NAR* loci: *DIAGEOTROPICA* (3 alleles), *IAA1a* (6), *IAA1b* (4), and *IAA2* (5) [11,23]. Twelve of the remaining 39 mutants with the strongest growth phenotypes were selected for whole-genome sequencing. The sequenced genomes were scanned for polymorphisms in auxin- or rhizoid-related genes and for genes with independent mutations in multiple mutant lines. Sequence analysis also revealed that the *DR5* element was tandemly repeated approximately 60 times. This is a possible explanation for the instability of the reporter.

We recovered 3 lines, *narD122*, *narD157*, and *narC18*, with lesions in the *Pp3c9_21330* locus. This locus encodes a clade-D ARF designated as *PpARFd1* (Fig 1B). Since these lines have not been backcrossed, they contain many mutations in addition to the lesions in *ARFd1*. To determine if the phenotype observed in these lines is due to the *arfd1* mutation, we deleted *ARFd1* in wild type (DR5:*DsRed2 Gd/NLS4)* and in the *narD122* line using CRISPR/Cas9 (Fig 1A). Lines with deletions of *ARFd1* were identified in both backgrounds (S1 Fig). Wild-type plants produce rhizoid-like filaments in place of phyllids (leaf-like structures) on gametophores when grown on medium containing 5 μM NAA. In contrast, *narD122*, *narD122/ Δarfd1*, and *Δarfd1* plants all produced some green phyllids consistent with a reduction in auxin response (Fig 1A, black arrowheads). Because *Δarfd1* plants had comparable phenotypes independent of genetic background, we concluded that the auxin-related phenotype in *narD122* is due to a defect in *ARFd1*.

## Clade-D ARFs are required for a robust auxin response

The phenotype of the *Δarfd1* mutant is mild compared to previously recovered *nar* mutants [11,24]. However, there are 2 ARFd genes in *P. patens*, *ARFd1* and *ARFd2* (*Pp3c15_9710*). To determine the contribution of *ARFd2* to the auxin response, we used CRISPR/Cas9 to delete *ARFd2* in the *DR5* background. For this experiment, the DR5:*DsRed2 Gd/NLS4* [18] had been backcrossed to the Reute ecotype [17]. The 35S:NLS-GFP-GUS cassette was not selected for and thus absent in the progeny [18,25]. The *DR5*-carrying progeny was backcrossed to Reute 2 additional times. For the remainder of the manuscript, this *DR5* line is the only one used. After recovering the *Δarfd2-8* allele, we used CRISPR/Cas9 to delete *ARFd1* from both *Δarfd2* and the same *DR5* background. In this way, we generated single *Δarfd1-18* and *Δarfd2-8* mutants as well as the double knockout (Fig 2 and S1 Fig). For simplicity, the 2 recovered ARFd double mutants are referred to as *Δarfd^{dub}*#1 and #3. Where appropriate, we make comparisons using both the *DR5* line and the Reute wild type.

Previous reports demonstrated that auxin treatment results in down-regulation of genes involved in chloroplast function and photosynthesis [18]. Indeed, plants lacking *Aux/IAA* transcriptional repressors, and thus constitutively executing the auxin response, have less chlorophyll than wild type [18]. Therefore, to assess the level of auxin signaling in the clade-D ARF mutants, we measured the chlorophyll content of plants grown on media with or without NAA (Fig 2). Because of the 21-day duration of the growth assay, we used NAA to avoid the photodegradation that can occur with IAA. Nitrogen levels in the medium can also affect chlorophyll levels (S2A and S2B Fig). Ammonium-deficient (BCD) medium with NAA supplementation has an additive effect in reducing the chlorophyll content. Chlorophyll content in wild-type and *DR5* lines on BCD plus NAA is at or near the detection limit, making comparisons difficult to interpret. Therefore, to accurately measure chlorophyll content, we spot-inoculated moss tissue onto ammonium-supplemented BCDAT medium with or without NAA (Fig 2). After 3 weeks on BCDAT + 0.5 μM NAA, chlorophyll levels on Reute wild-type plants were 43.7% of those grown without NAA. Similarly, the *DR5* plants had 41.9% as much chlorophyll on 0.5 μM NAA as the untreated controls (Fig 2B). On 5 μM NAA, wild-type and *DR5*

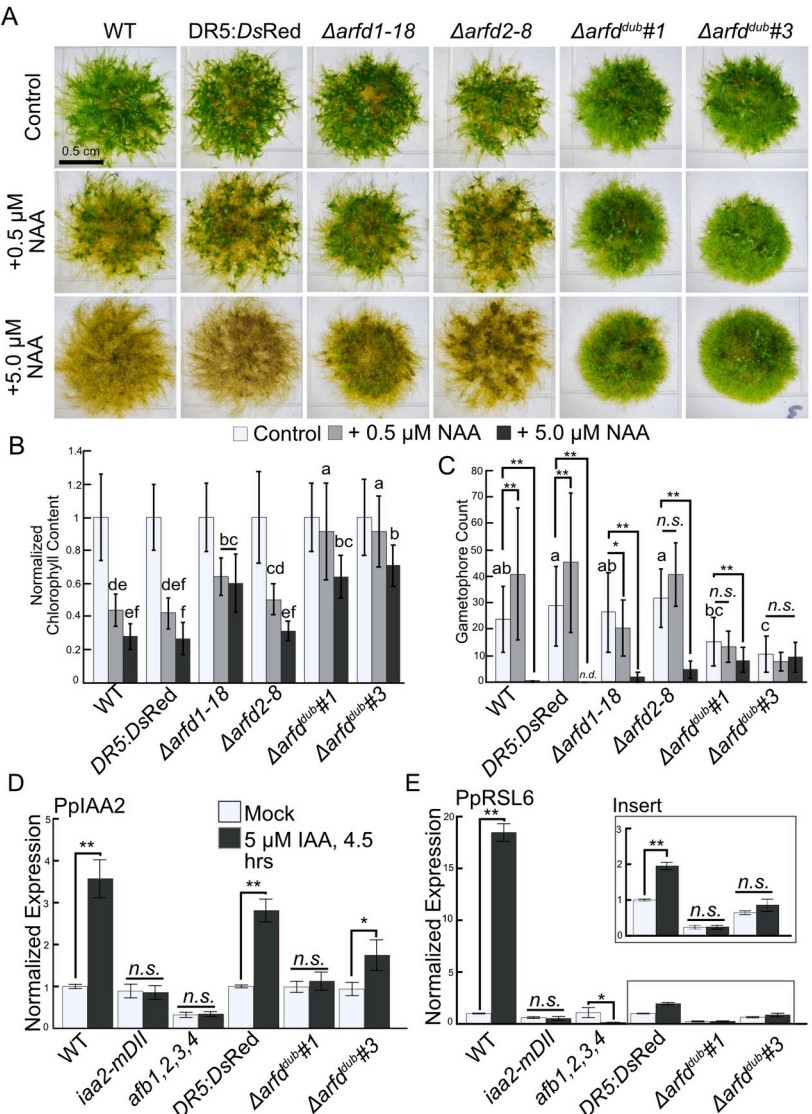

**Fig 2. Deletion of *ARFd* genes results in changes in morphology and auxin resistance.** (A) Images of 21-day-old colonies grown from inocula on indicated medium. (B) Chlorophyll content of 21-day-old colonies grown on indicated media, normalized to chlorophyll content on BCDAT. *n* = 18 colonies across 3 replicates. Error bars are standard deviation. Letters indicate statistical groups as determined by a TukeyHSD post hoc test of an ANOVA. (C) Average gametophore count of 21-day-old colonies grown on indicated media. *n* = 18 colonies across 3 replicates. Error bars are standard deviation. Letters indicate statistical groups as determined by a TukeyHSD post hoc test of an ANOVA. ** = *p*-value <0.001. (D) qPCR data showing auxin-induced changes in gene expression of *PpIAA2*. (E) qPCR data showing auxin-induced changes in gene expression of *PpRSL6*. Error bars are SEM. * = *p*-value <0.05, ** = *p*-value <0.001, n.s. = not significant. The underlying data for panels B, C, D, and E are in S1 Data. ARF, auxin response factor; NAA, 1-naphthaleneacetic acid.

had 28% and 26.6% of the chlorophyll, respectively, as the no auxin control. In contrast, the effect of NAA on the *Δarfd*^*dub*^ line was much less than the wild type (64.1% and 71.0% of control, respectively, on 5 μM NAA; Fig 2B). *Δarfd1* plants on 5 μM NAA have 60.4% as much chlorophyll content compared to no NAA. In contrast, *Δarfd2* plants grown on 5 μM NAA have 69% less chlorophyll than the mock-treated counterparts. Indeed, the auxin response of *Δarfd2* plants groups statistically with wild type and *DR5*. This difference suggests that ARFd1

makes a larger contribution to the auxin response than ARFd2. Therefore, we focused some subsequent experiments on *ARFd1*.

In addition to affecting chlorophyll levels, previous work has demonstrated a link between auxin and gametophore development. Treating *P. patens* with the IAA biosynthesis inhibitor L-Kynurenine (L-Kyn) reduces auxin levels, resulting in small plants with few gametophores [18,26]. Additionally, *P. patens* mutants with reduced auxin signaling via up-regulation of a clade-B repressing ARF [14] or stabilized Aux/IAAs [11] produce few gametophores. Therefore, to corroborate that auxin signaling is reduced in the *arfd* lines, we counted gametophores in the presence or absence of NAA (Fig 2C and S2C Fig). Both wild type and *DR5* produced more gametophores on BCDAT supplemented with 0.5 μM NAA than on control plates (Fig 2C), and 5 μM NAA, however, had an inhibitory effect on gametophore development. While wild-type plants initiate gametophore-like structures on 5 μM NAA, these structures develop rhizoids instead of phyllids [18]. For these experiments, we did not count these structures. On control medium, both *Δarfd1-18* and *Δarfd2-8* produced near-wild-type numbers of gametophores. Both single mutants had a mild response to 0.5 μM NAA and a strong response to 5.0 μM NAA. Interestingly, and in contrast to wild-type plants, both single mutants produced at least a few gametophores on 5 μM NAA. *Δarfd^dub* plants, however, produce fewer gametophores on unsupplemented BCDAT than wild type or the single mutants. This suggests that at least one functional ARFd is required for proper gametophore initiation. Consistent with decreased auxin signaling in *Δarfd^dub*, NAA treatment has a minimal effect on gametophore production in these mutants.

The diminished physiological and developmental response to exogenous NAA in *Δarfd^dub* plants suggests that the ARFds function as transcriptional activators. To obtain more direct evidence for this hypothesis, we analyzed the expression of auxin-regulated genes in the *Δarfd^dub* lines using qRT-PCR. For comparison, we used 2 lines with diminished auxin signaling—an IAA2-stablized line (*iaa2-mDII*) and a line lacking all 4 *AUXIN F-BOX* genes (*afb1,2,3,4*) (S1B and S1C Fig). We treated each line with 5 μM IAA (or mock) for 4.5 h. Finally, we normalized changes in gene expression to the parental line for each mutant (Fig 2C). Neither of the two auxin-induced genes, *IAA2* and *RSL6*, were up-regulated in the *iaa2-mDII* or *afb1,2,3,4* line and there was only nominal up-regulation in the *Δarfd^dub* lines. Taken together, both the physiological, developmental, and gene expression data point to a role for clade-D ARFs as transcriptional activators in *P. patens*.

## The PB1 domain is required for ARFd1 function

Our results indicate that the clade-D ARFs are positive regulators of auxin signaling despite the fact that they lack a DBD and presumably cannot bind DNA directly. To gain insight into how the ARFds regulate transcription, we utilized the *arfd1^T653L* mutation identified in the *narD122* line (Fig 1). To confirm that the *arfd1^T653L* substitution disrupts auxin signaling, we employed CRISPR/Cas9 in combination with an oligonucleotide carrying the mutation to recreate the T653L substitution in a *pARFd1:ARFd1-mYPet/Δarfd2/DR5:DsRed* (*ARFd1-mYPet*) background. We refer to this line as *arfd1^T653L-mYPet*. Similarly, we generated lines containing a Y643* mutation (*arfd1^Y643*-mYPet*), which generates a premature stop codon just before the PB1 domain (S3A and S3B Fig). Interestingly, we consistently observed that arfd1^T653L-mYPet signal was brighter than ARFd1-mYPet (S3C and S3D Fig). We determined the effect of these mutations on auxin signaling by measuring the chlorophyll content of tissue grown on medium with or without 5 μM NAA for 21 days (Fig 3A and 3B). Tissue of the control (*DR5*) and *ARFd1-mYPet* lines had 37% as much chlorophyll on mock medium compared to NAA-supplemented medium. Similar to the results presented in Fig 2, chlorophyll levels in the

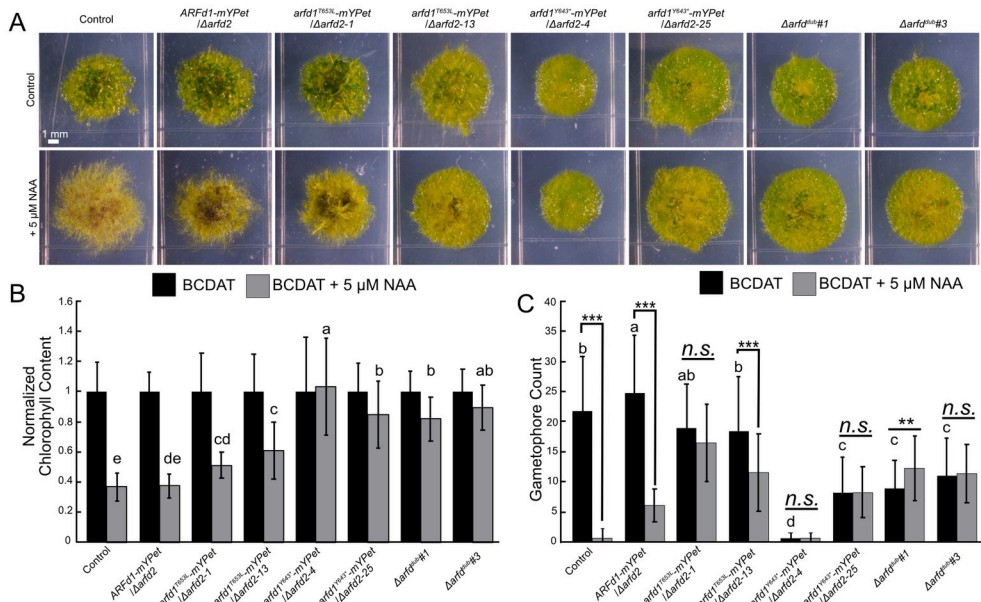

**Fig 3. ARFd1 function depends on the PB1 domain.** (A) The 21-day-old colonies of ARFd1 mutants grown on BCDAT +/- 5 μM NAA. Control line is *DR5:DsRed* parental line. (B) Chlorophyll content of whole colonies imaged ($n \geq 18$ colonies across 3 replicates for each line and treatment). Error bars are standard deviation. Letters indicate statistical groups as determined by a TukeyHSD post hoc test of an ANOVA. (C) Average gametophore count for $n \geq 18$ colonies across 3 replicates grown BCDAT +/- 5 μM NAA for 21 days. Letters indicate statistical groups determined by a TukeyHSD post hoc test of an ANOVA. ** = $p < 0.01$, *** = $p < 0.001$. The underlying data for panels B and C are in S1 Data. ARF, auxin response factor; NAA, 1-naphthaleneacetic acid.

*Δarfd<sup>dub</sup>* lines were only slightly affected by NAA treatment. Likewise, independent *arfd1<sup>Y643*</sup>-mYPet* lines displayed a similar level of NAA resistance indicating that the PB1 domain is necessary for ARFd1 function.

Independent *arfd1<sup>T653L</sup>-mYPet* lines exhibit an auxin sensitivity that is intermediate between the control and *Δarfd<sup>dub</sup>* lines. Therefore, we conclude that the arfd1<sup>T653L</sup> protein retains some activity. The *arfd1<sup>T653L</sup>* lines exhibit intermediate gametophore development phenotypes as well. They develop gametophores at levels comparable to control plants (Fig 3C), but interestingly gametophore number is only minimally affected by NAA treatment (Fig 3A and 3C). These observations are consistent with published findings of auxin-insensitive *Physcomitrium* mutants [11,14]

## The ARFd proteins have a role in the chloronema–caulonema transition in an ammonium-dependent manner

Previous studies have shown that auxin promotes the chloronema–caulonema transition. Consistent with this, we found that *Δarfd<sup>dub</sup>* colonies are noticeably round when growing on ammonium-supplemented medium (BCDAT). These observations suggest that the ARFd proteins are required for the development of faster-growing caulonemata. However, on ammonium-deficient medium (BCD), we do not detect a difference in colony circularity (S2D Fig). To address this issue directly, we regenerated plants from single protoplasts on both ammonium-supplemented and ammonium-deficient media (Fig 4A) and determined the extent of caulonemal development by measuring cell cross-wall angle. The angle of the cross-wall is widely recognized as a key morphological distinction between chloronemata and caulonemata.

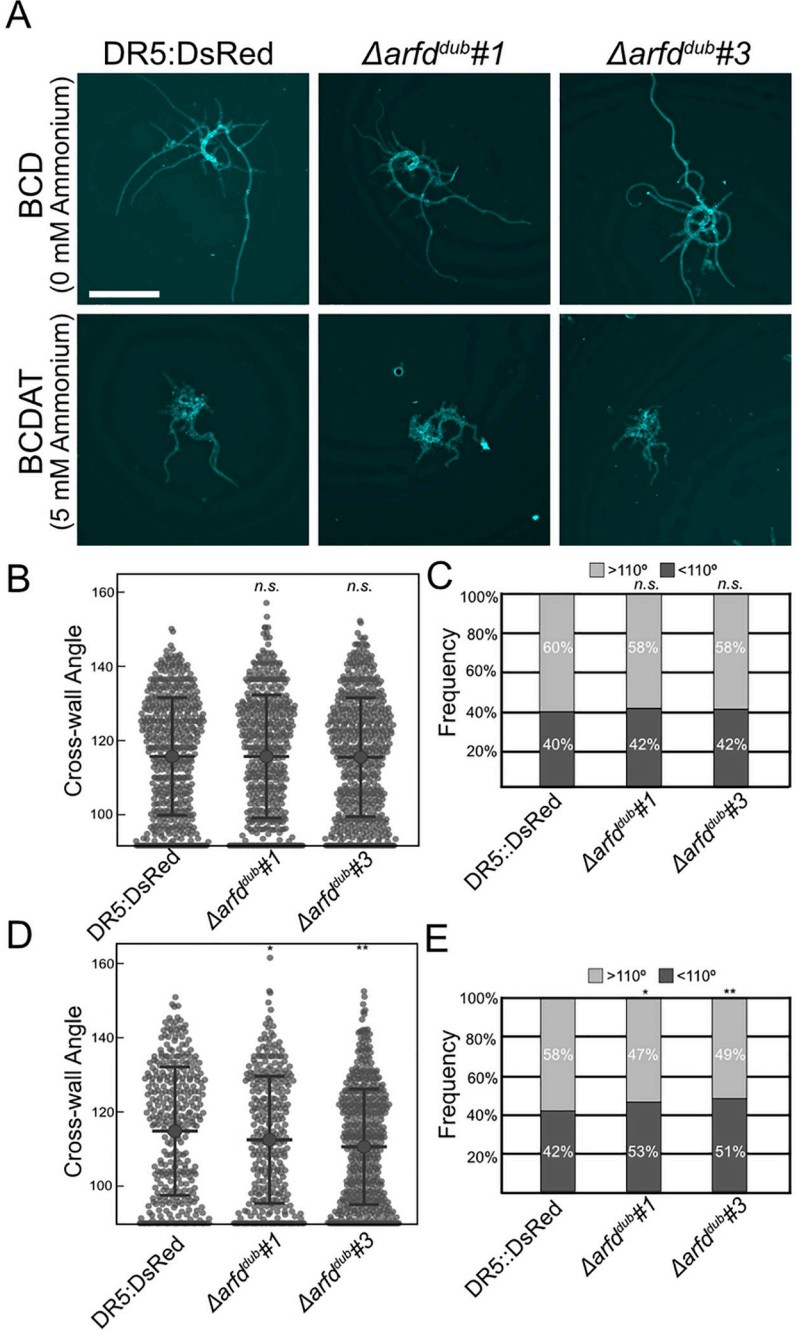

**Fig 4. *Δ*arfd^dub lines display an ammonium-dependent delay in caulonemal development.** (A) Images of representative 7-day-old plants grown from regenerated protoplasts of indicated line grown on medium with or without ammonium. Plants stained with Calcofluor for imaging, scale bar is 500 μm. (B) Protonemal cross-wall measurements from *DR5*:DsRed ($n = 532$), *Δarfd^dub*#1 ($n = 502$), and *Δarfd^dub*#3 ($n = 578$) plants grown on BCD from at least 2 biological replicates. The *Δarfd^dub* cross-walls were not significantly different from *DR5*:DsRed (Student's *T* test). (C) Frequency of cross-walls from BCD-grown plants binned as >110˚ or <110˚. The distribution of each line was the same (Kolmogorov–Smirnov, K.S., test). (D) Protonemal cross-wall measurements of *DR5*:DsRed ($n = 538$), *Δarfd^dub*#1 ($n = 335$), and *Δarfd^dub*#3 ($n = 526$) plants grown on BCDAT medium from 2 biological replicates. Both *Δarfd^dub* lines are enriched for low-angle cross-walls (Student's *T* test, * = $p < 0.05$, ** = $p < 0.001$). (E) Frequency of cross-walls from BCDAT-grown plants binned as >110˚ or <110˚. Both *Δarfd^dub* lines have more chloronemata (Kolmogorov–Smirnov test, * = $p < 0.05$, ** = $p < 0.001$). Error bars are standard deviation. The underlying data for panels B, C, D, and E are in S1 Data. Representitive images demonstrating cross-wall image analaysis are in S2 Data. ARF, auxin response factor.

Chloronemata have cross-walls that are perpendicular to the growth axis of the filament, while caulonemata cross-walls are oblique. The wild-type *DR5* line on BCD had an average cross-wall angle of 114.1 ± 15.9˚ (Fig 4B). Consistent with the morphological distinction between chloronemata and caulonemata, the cross-wall data form a bimodal distribution around 110˚. Therefore, we used 110˚ as our cutoff to differentiate chloronemata and caulonemata in our data. In wild-type *DR5* lines, 40% of measured cross-walls fell below this threshold and were scored as chloronemata, while 60% of cross-walls belonged to caulonemata (Fig 4C). Interestingly, both *Δarfd^{dub}* mutants have nearly the same cross-wall average (114.1 ± 16.6˚ and 113.9 ± 16.1˚, respectively) and only slight differences in distribution (42% chloronema, 58% caulonemata for each) compared to the parental *DR5* line (Fig 4B and 4C). Neither T-tests of the averages and Kolmogorov–Smirnov test of the distributions yielded a significant difference. Taken together, these data suggest that *Δarfd^{dub}* plants have a wild-type ability to make caulonemata on ammonium-deficient BCD medium.

In contrast, the presence of ammonium in the medium had a significant effect on caulonemal development. In these conditions, the average cross-wall angle for *DR5* plants was 114.8 ± 17.3˚, with 42% of measured cells counting as chloronema (Fig 4D and 4E). Meanwhile, both *Δarfd^{dub}* lines had a slight, but significant, enrichment in chloronemata. This was true both in the average cell wall angle (112.5 ± 17.1˚ and 110.6 ± 15.6˚, respectively) and the percentage of cells with cross walls less than 110˚ (53% and 51%, respectively). Conducting a Kolmogorov–Smirnov test on the distribution of cross-wall data determined that the detected shift was significant.

To assay auxin signaling between the 2 media types, we conducted qPCR on RNA extracted from wild type grown on BCD or BCDAT for 3 days. *RSL6*, which is strongly induced by IAA treatment, has a 4-fold higher expression level in tissue grown on BCD (S2E Fig). This suggests a general increase in auxin signaling. Since the *Δarfd^{dub}* phenotype is strongest on BCDAT, and therefore required for proper development in those conditions, one possibility would be that clade-D ARFs are up-regulated on BCDAT. Counter to this expectation however, we did not detect a significant difference in *ARFd1* expression level in plants grown on BCD or BCDAT (S2E Fig).

Given that the chloronemal enrichment was slight in *Δarfd^{dub}*, we hypothesized that ammonium-dependent delay in caulonemal differentiation should be more pronounced in plants with more severe mutations in the auxin-signaling pathway. Therefore, we grew *iaa2-mDII* and *afb1,2,3,4* plants from regenerated protoplasts on medium with or without ammonium. Independent of ammonium levels, both *iaa2-mDII* and *afb1,2,3,4* protonemata are enriched in chloronemata (S4 Fig). These observations are consistent with previous reports that auxin signaling is required for caulonemata development [7,11]. For each mutant, the ammonium-dependent chloronemal enrichment was enhanced compared to wild type. In the absence of robust auxin signaling, the contribution of ammonium sensing to the chloronemal-caulonemal transition is more apparent.

Taken together, auxin signaling and nutrient sensing both promote the chloronema–caulonema transition. ARFds are required for caulonemal development, but only in nutrient-rich conditions.

## Clade-D ARFs are required for proper branch filament initiation

Thus far, we have demonstrated that clade-D ARFs are required for a robust auxin response and for caulonemal development on medium containing ammonium. Based on these results, we hypothesized that clade-D ARFs may be necessary when high levels of auxin signaling are required. Foundational work in the moss *Funaria* suggests that IAA regulates the number of

filament branches [15], as *Funaria* protonemata grown in medium supplemented with IAA developed more branches. However, more recent findings in *Physcomitrium* suggest that auxin has little effect on the number of branches per filament [7]. In our conditions, we observed increased *DR5* activity in young filament branches (Fig 5A), suggesting that auxin signaling may play a role in protonemal branching. Therefore, we compared filament branch emergence frequency in the *DR5* and *arfd* lines. We scored each filament by the position of the first branch relative to the apical cell. Filaments from the *DR5* control line typically branch at the second subapical cell (68.4%) (Fig 5B and 5H and S5A Fig). Less often, the first subapical cell (9.3%) or third subapical cell (17.3%) will branch first (Fig 5H and S5A Fig). Exposure to 5 µM IAA for 3 days induces premature branch emergence in the *DR5* line with an increase in branch formation on the apical (3.6%) and first subapical (39.6%) cell (Fig 5E and 5H and S5A Fig). This observation supports a model in which the level of auxin signaling contributes to the timing of branch initiation.

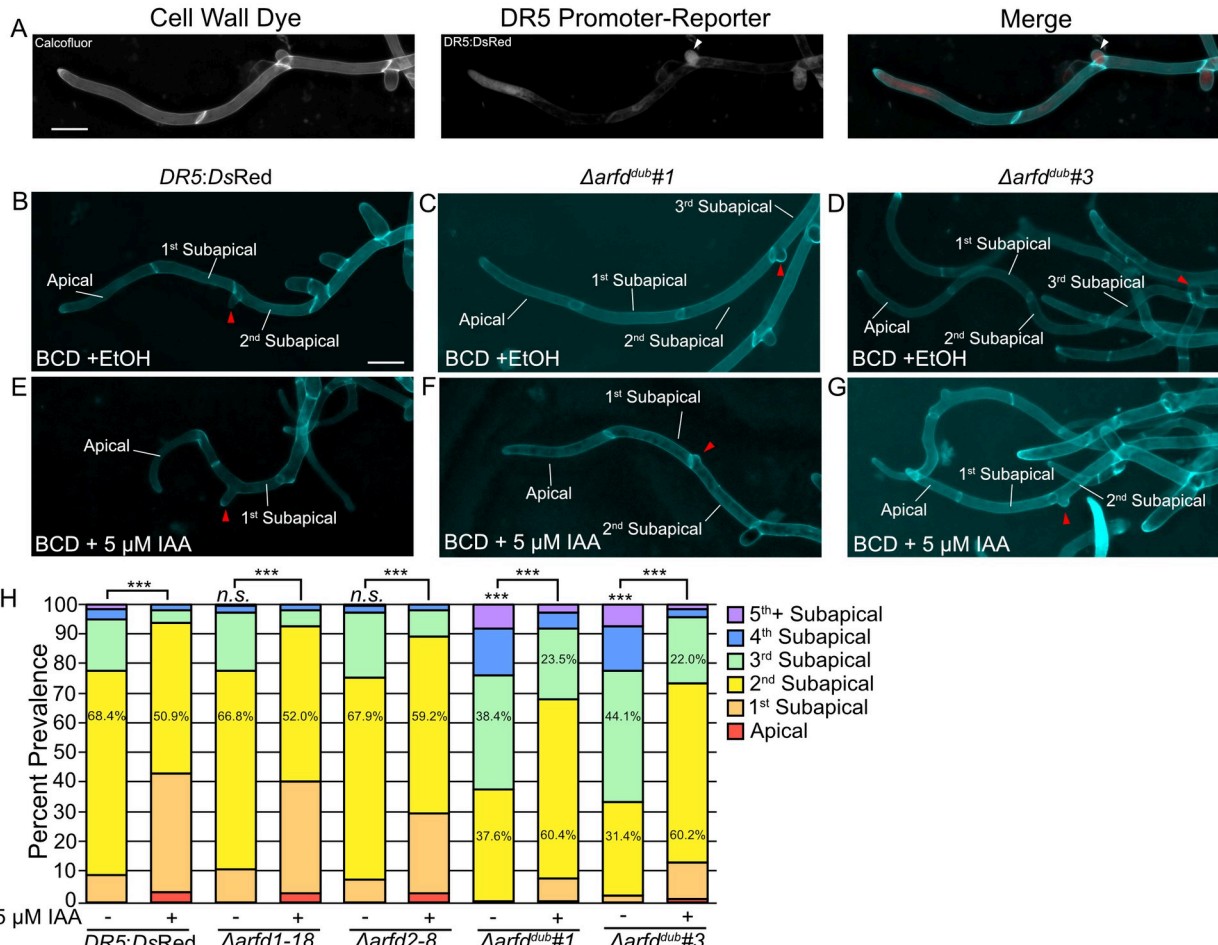

**Fig 5. Branch initiation is delayed in *Δarfd*^*dub* mutants.** (A) Representative micrographs of the *DR5* line showing that auxin signaling increases in filament branch cells (white arrowhead). Cell wall stained with calcofluor. Scale bar is 50 µm. (B–D) Representative micrographs of indicated lines grown on BCD for 3 days stained with calcofluor. Red arrowhead highlights emerging branch. Scale bar is 50 µm. (E–G) Representative micrographs of indicated lines grown on BCD + 5 µM IAA for 4 days. Scale bar is 50 µm. (H) Frequency of branching in control and *arfd* mutants grown on medium with or without 5 µM IAA. Distribution of branching is significantly different for each line (K.S. test, *** = $p < 0.001$). The branching pattern of *Δarfd1-18* and *Δarfd2-8* are not significantly different from *DR5* line. Both *arfd*^*dub* lines produce filaments that exhibit a branching delay. Statistics for branching determined by assigning numerical values to cell number (apical = 0, subapical = 1, etc.) and performing K.S. test. The underlying data for panel H are in S1 Data. ARF, auxin response factor; IAA, indole acetic acid.

Both *Δarfd1-18* and *Δarfd2-8* had only slight and statistically insignificant changes in branch timing compared to control (Fig 5H and S5A Fig). In contrast, both *Δarfd$^{dub}$* lines exhibit a delay in branch emergence (Fig 5C, 5D and 5H and S5A Fig). While both double mutant lines frequently branch at the second subapical cell (37.6% and 31.4%, respectively), there is an increase in the number of late-forming branches (Fig 5H and S5A Fig). Indeed, the plurality of *Δarfd$^{dub}$* filaments formed the first branch on the third subapical cell (38.4% and 44.1%, respectively) compared to 17.3% in the *DR5* line. Additionally, 7% to 8% of mutant filaments had branches emerging on or after the fifth subapical cell, compared to only 1.3% of filaments in the *DR5* line. Similar to *DR5* plants, we observed a stimulation of branch emergence in *Δarfd$^{dub}$* lines grown on 5 μM IAA (Fig 5F, 5G and 5H and S5A Fig). On auxin, the frequency of branching at the second subapical cell increased to 60.4% and 60.2% for the double mutant lines, respectively. Additionally, branching on or after the fifth subapical cell occurred in <3% of mutant filaments when grown on auxin. However, unlike auxin-treated wild-type plants, there are relatively few branches on the apical (<1.5%) or first subapical cell (<12.5%) after auxin treatment of *Δarfd$^{dub}$* plants. These results show that auxin signaling is a major factor in determining filament branch timing and that the ARFd proteins contribute to this process.

These results indicate that auxin plays a fundamental role in protonemal filament branching. We sought to validate our studies by measuring branch emergence in other auxin signaling mutants. We found that both the *iaa2-mDII* and *afb1,2,3,4* lines exhibit a strong delay in filament branching (S5B, S5C and S5D Fig), confirming a role for auxin in branch formation.

## ARFd1 is expressed in protonemal filaments and the base of growing gametophores

Since ARFd1 is involved in the growth and development of protonemata and gametophores, we used the *ARFd1-mYPet* line to assess the pattern of *ARFd1* expression (Fig 6). Given the role of ARFds in the chloronemal-caulonemal transition, we hypothesized ARFd1 would be enriched in a specific cell type. To this end, we measured mYPet signal in 7-day-old *ARFd1-mYPet* plants regenerated from protoplasts and the angle of the proximal cell wall (Fig 6A). ARFd1-mYPet signal was not significantly different across cell type using the same 110° delineation between chloronemata and caulonemata (Fig 6B).

The *Δarfd$^{dub}$* lines exhibit delayed gametophore development (Fig 3C). When we examined *ARFd1-mYPet* tissue 10 days after homogenization, we consistently detected a bright ARFd1-mYPet signal in the protonemal cells at the base of the developing gametophore (Fig 6C, white arrowhead). This observation is consistent with an important role for clade-D ARFs in gametophore initiation. Later in gametophore development, we detected ARFd1-mYPet at the base and in rhizoids (Fig 6D), suggesting that ARFd1 is involved in gametophore bud initiation, and potentially in rhizoid development.

## ARFd1 function requires oligomerization

One important question is how clade-D ARFs activate gene expression without a DBD. One hypothesis is that these ARFs are recruited to the DNA via an interaction with canonical ARFs. Most ARFs have a PB1 dimerization domain, and because T653 is in the PB1 domain of ARFd1 (Fig 1B), this mutation could disrupt the ability of ARFd1 to interact with canonical ARFs. To test this possibility, we conducted yeast-2-hybrid assays with full-length ARFs to assess their ability to interact with ARFd1 (Fig 7A). We found that ARFd1 interacts with the clade-A protein ARFa8. Conversely, there is minimal interaction with ARFb1, a repressing clade-B ARF. Importantly, full-length *arfd1$^{T653L}$* also interacted with ARFa8, suggesting that

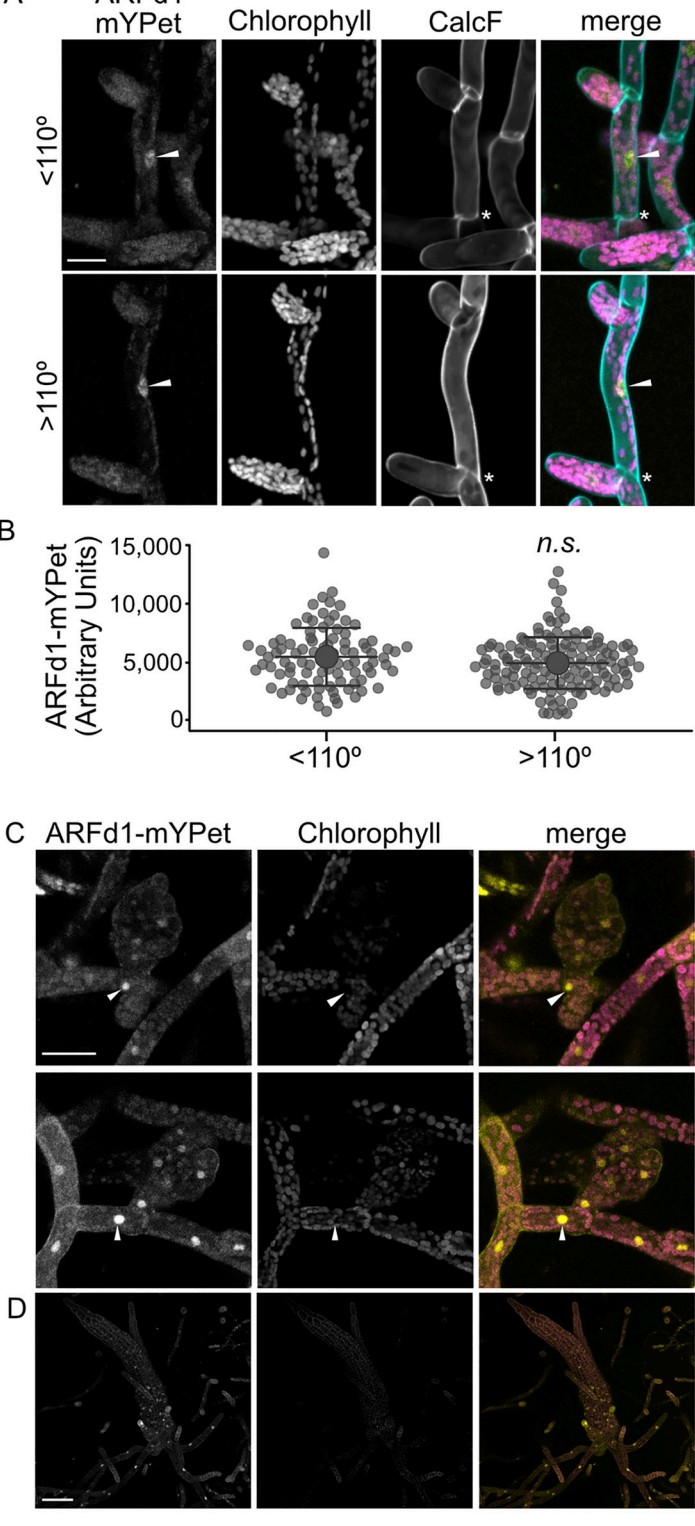

**Fig 6. *ARFd1* expression pattern.** (A) Representative micrographs from 7-day-old plants regenerated from *ARFd1-mYPet* plants stained with calcofluor. Top is a cell with the proximal cell wall (asterisk) of <110˚ (chloronema cell), bottom is >110˚ (caulonema cell). White arrowhead highlights nuclear ARFd1mYPet signal. Scale bar is 25 μm. (B) Nuclear ARFd1-mYPet signal binned by cell wall angle (*n* = 299 cells across 4 biological replicates, error bars indicate standard deviation, statistics determined with Student's *T* test). (C) Representative micrographs of young

gametophore buds. ARFd1-mYPet is enriched in the protonemal cell at the base of the developing gametophore (white arrowheads). Scale bar is 50 μm. (D) ARFd1-mYPet signal is highest at the base of in maturing gametophores. Scale bar is 100 μm. The underlying data for panel B are in S1 Data. Representitive images describing nuclear signal image analaysis are in S2 Data. ARF, auxin response factor.

the mutation does not eliminate the ability of ARFd1 to dimerize with other ARFs in yeast cells.

PB1 domains consist of separate positively and negatively charged surfaces (formerly DIII and DIV, respectively) that facilitate oligomerization through surface charge interactions in a head to tail fashion [5,6]. For ease of discussion, we will refer to the 2 faces as DIII and DIV. The T653L mutation affects the DIII face of ARFd1 (Fig 7A), and therefore may affect the interaction with the DIV of other ARFs. To test this possibility, we made ΔDIV versions of ARFd1 and arfd1$^{T653L}$ and performed yeast-2-hybrid assays. ARFd1ΔDIV retains the ability to interact with ARFa8. However, arfd1$^{T653L}$ΔDIV cannot interact with ARFa8 indicating that the DIV domain in arfd1$^{T653L}$ is responsible for its interaction with ARFa8 in yeast. We verified this result in plants using split yellow fluorescent protein (YFP) assays in *P. patens* protoplasts. Interestingly, we observed interaction as subnuclear puncta (Fig 7B).

We also tested the ability of these proteins to interact with PpIAA1a, as Aux/IAAs also contain PB1 domains. While full-length arfd1$^{T653L}$ interacted with PpIAA1a as well as full-length ARFd1 in both yeast and in split YFP experiments, arfd1$^{T653L}$ΔDIV did not (Fig 7A). As a control, we tested the ability of the arfd1$^{Y643*}$ protein to interact with ARFa8 and IAA1a. As expected, arfd1$^{Y643*}$ did not interact with either protein. This confirms that ARFd1 interacts with other ARFs and Aux/IAAs solely through the PB1 domain.

Taken together, these results indicate that the T653L mutation disrupts the function of one of the 2 charged faces in the PB1 domain. The remaining face (DIV) is sufficient to enable interaction with ARFa8 and IAA1a and partial ARFd1 function in plants. The fact that both faces are required for full activity strongly suggests that ARFd oligomerization contributes to its activity.

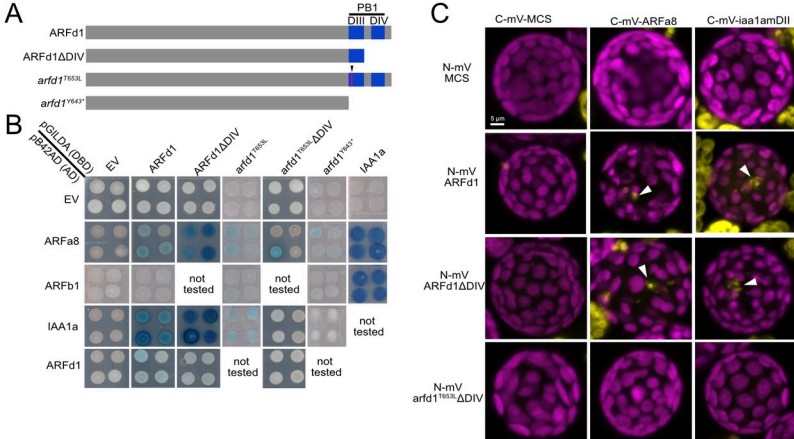

**Fig 7. PpARFd1 requires oligomerization for full activity.** (A) Wild-type and mutant ARFd1 constructs used for yeast-2-hybrid assay. (B) Yeast-2-hybrid results. ARFd1 interacts with both the activating ARFa8 as well as the PpIAA1a transcriptional repressor. Arfd1$^{T653L}$ cannot interact without the negatively charged DIV domain. (C) Split YFP assay confirming that ARFd1-IAA1a and ARFd1-ARFa8 interaction is abolished when arfd1$^{T653L}$ protein can only dimerize through DIII. ARFd1 interaction with both IAA1a and ARFa8 occurs in puncta (white arrowheads). ARF, auxin response factor; IAA, indole acetic acid; YFP, yellow fluorescent protein.

## The middle region of ARFd1 is required for function

Our interaction and genetic data strongly suggests ARFd1 functions via oligomerization of clade-A ARFs via the PB1 domain. However, these data do not rule out the possibility that clade-D ARFs function to titrate the Aux/IAA repressors away from auxin-responsive promoters [20]. Indeed, the 2 models, activation and titration, are not necessarily mutually exclusive. However, if clade-D ARFs function primarily by titrating Aux/IAAs, the intrinsically disordered MR would presumably have a minor role. To test this hypothesis, we expressed mutant versions of ARFd1 under control of the maize ubiquitin promoter, stably integrated at the *P. PATENS INTER-GENIC1 (PIG1)* locus [27]. We recovered 2 stable transgenic lines for each construct for subsequent testing.

As a control, we tested full-length mYPet-ARFd1 also under control of the maize ubiquitin promoter (Fig 8A). The *pUbi:mYPet-ARFd1* lines were hypersensitive in terms of chlorophyll

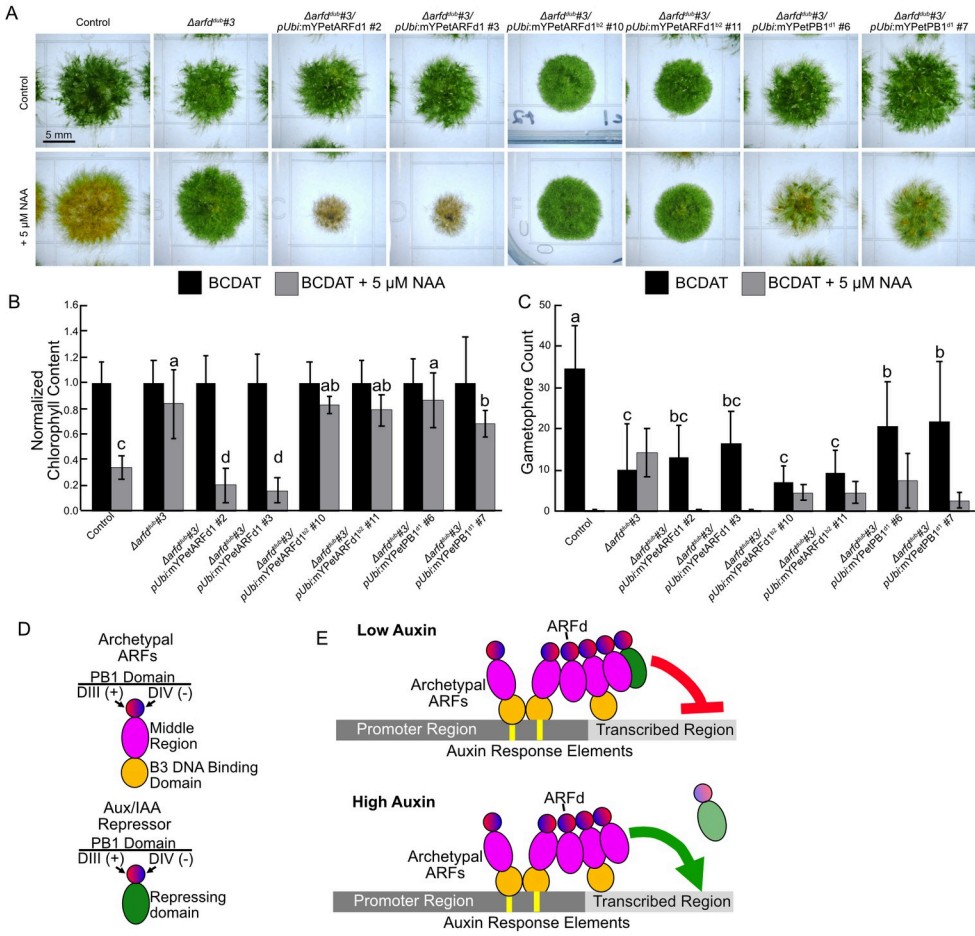

**Fig 8. A model for clade-D ARF function.** (A) Representative micrographs of 21-day-old spot inocula on BCDAT +/- 5 μM NAA. (B) Chlorophyll content of 21-day-old colonies, normalized to chlorophyll content on BCDAT. $n \geq 18$ colonies across 3 replicates. Error bars are standard deviation. Letters indicate statistical groups as determined by a TukeyHSD post hoc test of an ANOVA. (C) Gametophore counts of 21-day-old inocula. Each line produced a significantly different number of gametophores with NAA treatment (Student's *T* test, $p \leq 0.05$). (D) Schematic of the domains of ARFs and Aux/IAA Repressors. (E) In low auxin conditions, ARFds oligomerize with canonical ARFs as well as Aux/IAAs to an auxin-inducible gene. Once the Aux/IAA is removed in an auxin-dependent manner, clade-D ARFs activate gene expression in conjunction with archetypal clade-A ARFs. The underlying data for panels B and C are in S1 Data. ARF, auxin response factor; IAA, indole acetic acid; NAA, 1-naphthaleneacetic acid.

content and gametophore production when challenged with 5 μM NAA for 21 days, supporting the hypothesis that ARFd1 activates transcription (Fig 8B and 8C). Interestingly, *pUbi: mYPet-ARFd1* was unable to restore wild-type numbers of gametophores on BCDAT medium. Taken together with our expression analysis (Fig 6), this suggests that *ARFd1* may need to be expressed with a specific expression pattern to trigger gametophore development. Next, we assessed the role of the MR in ARFd1 function. We replaced the region of ARFd1 between S252 and P496 with 231AA of PpARFb2's MR (ARFd1[b2]). Notably, this region includes the B3 repression domain (BRD) motif KLFG. BRD-like motifs facilitate the interaction between ARFs and the co-repressor TOPLESS [28]. When treated with 5 μM NAA, the *pUbi:mYPet-ARFd1[b2]* lines displayed a similar reduction in chlorophyll level as *Δarfd[dub]*#3. Likewise, the number of gametophores that developed on *pUbi:mYPet-ARFd1[b2]* colonies did not change in response to NAA. These results indicate that MR of ARFd1 is required for its function. Finally, to determine if the PB1 domain of ARFd1 is sufficient to restore auxin signaling to *Δarfd[dub]* plants, we expressed the C-terminal 265AA of ARFd1 (PB1[d1]). This region is downstream of the intrinsically disordered MR and contains the PB1 domain. If the primary function of clade-D ARFs is to titrate the Aux/IAAs away from the promoter region, we would expect *pUbi:mYPet-PB1[d1]* to phenocopy *pUbi:mYPet-ARFd1*. Instead, the effect of NAA on chlorophyll level in these lines was similar to the *pUbi:mYPet-ARFd1[b2]* lines. While *pUbi:mYPet-PB1[d1]* lines produce more gametophores than the other transgenic lines, they did not develop as many as wild type. Taken together, our data strongly suggest that clade-D ARFs function by activation after oligomerization with clade-A ARFs rather than by titration of Aux/IAA repressors.

## Discussion

Unlike other members of the ARF family of transcription factors, the clade-D ARFs lack a DBD. Previous work with another bryophyte, *M. polymorpha*, found that loss of the single clade-D ARF results in effects on auxin-regulated transcription [20]. However, the role clade-D ARFs play in plant development, as well as how they affect gene expression, were open questions. Our experiments demonstrate that clade-D ARFs function as transcriptional activators and have an important role in the progression of caulonema and gametophore development in *P. patens.*

Clade-D ARFs have a clear role in *P. patens* development because *Δarfd[dub]* colonies have a higher circularity than wild type. Previous work has associated high colony circularity with decreased auxin signaling. An increase in colony circularity has been interpreted as the reduced ability to make caulonemata [11,14,18]. When we directly examined protonemal development using regenerated protoplasts, we were surprised to find that *Δarfd[dub]* plants can make wild-type levels of caulonemata on ammonium-deficient BCD medium. However, *Δarfd[dub]* plants grown on ammonium-supplemented BCDAT medium exhibit a delay in caulonemal development. This suggests that a nutrient-sensing developmental pathway masks the developmental phenotypes of *Δarfd[dub]* in some conditions. Interestingly, even strong auxin-signaling mutants show a nutrient-dependent chloronema–caulonema transition. Future work will determine if the nutrient-dependent pathway interacts with the auxin-signaling pathway, or if it is a parallel genetic pathway in the chloronemal-caulonemal transition.

Independent of ammonium levels, protonemal branching is delayed in *Δarfd[dub]* plants. We observed a similar branching delay in strong auxin signaling mutants (*iaa2-mDII, afb1,2,3,4*). These data suggest that protonemal branch formation is an auxin-regulated process. These findings corroborate earlier studies in *Funaria* in which they found an increase in caulonemal

branches in response to auxin treatment [15]. In support of this hypothesis, exogenous IAA results in premature branching. In contrast, recent work using *pina,pinb* mutants and NAA treatment in *Physcomitrium* found no connection between auxin and caulonemal branches [7]. In our branching experiments, we did not differentiate chloronemata and caulonemata. Additionally, Thelander and colleagues measured branching as branches per cell and did not address the timing of branch emergence relative to the apical cell [7]. Taken together, we are confident that the timing of protonemal branching is informed by the level of auxin signaling. Additionally, clade-D ARFs are key regulators of this process.

In addition to the delay in protonemal branching, *Δarfd*[dub] plants exhibit delayed gametophore development. Interestingly, apart from this delay, *Δarfd*[dub] gametophores are phenotypically wild type. This suggests that the role of clade-D ARFs in gametophore development is restricted to initiation. In line with these observations, we detected an increase in ARFd1--mYPet levels in the protonemal cell at the base of young gametophore buds. While *Δarfd*[dub] plants have an ammonium-dependent delay in the chloronemal-caulonemal transition, we did not detect a significant difference in ARFd1-mYPet level between the 2 cell types regardless of ammonium levels in the medium. Additionally, ARFd1 transcript levels are unaffected by the ammonium levels in the 2 media types. In general, the *Δarfd*[dub] phenotypes are similar to those of the strong *iaa2-mDII* and *afb1,2,3,4* mutants, but slightly weaker. This leads us to propose that clade-D ARFs are involved in general auxin-mediated gene activation events (Fig 8).

ARFd1 activates gene expression despite lacking a DBD. Our data demonstrate that an intact PB1 domain is necessary, but not sufficient, for full function. PB1 domains facilitate interaction between ARFs and Aux/IAA repressors [29]. ARF–ARF interactions can occur via the PB1 domain [2,6]. Further, a recent report indicates that the DBD and PB1 domains act cooperatively to promote binding of ARF proteins to DNA [30]. Our studies indicate that ARFd1 interacts with both activating ARFa8, as well as PpIAA1a, presumably through the common PB1 domain. PB1 domains have 2 binding surfaces—positive and negative. By limiting protein–protein interactions to the positive face of ARFd1, we demonstrated that the arfd1[T653L] can dimerize but not oligomerize. The most likely model is that clade-D ARFs interact with DNA-bound clade-A activating ARFs (e.g., PpARFa8) and enhance transcriptional activation. Indeed, ectopic expression of chimeric ARFd1 with the BRD-containing MR of PpARFb2 does not restore auxin response to the *Δarfd*[dub] mutant. This indicates that the MR of clade-D ARFs is important for gene activation. Further, ectopic expression of the ARFd1 PB1 domain alone does not restore auxin response to *Δarfd*[dub] plants. *Arfd1*[T653L]/*Δarfd2* plants have intermediate phenotypes between *ARFd1/Δarfd2* and *Δarfd*[dub], suggesting that ARFd1-mediated transcriptional enhancement can occur through dimerization. However, for wild-type levels of auxin signaling ARFd1 has to multimerize with additional factors. One possible scenario is that ARFd1 recruits additional activating ARFs, either ARFa or ARFd, through their PB1 domains (Fig 8D and 8E). Our data also indicate that ARFd1 can bind the transcriptional repressor PpIAA1a, suggesting that ARFd1 function is auxin regulated via AFB-dependent degradation of the Aux/IAAs.

The work presented here demonstrates that clade-D ARFs are key players in bryophyte auxin signaling. Additionally, we demonstrate that ARFd oligomerization is essential for full function. Likewise, ARFd1 function depends on the MR, suggesting that clade-D ARFs do not activate gene expression solely by titrating Aux/IAA repressors. Although clade-D ARFs were lost in later-diverging plant lineages, it is tempting to speculate that clade-A ARF also form PB1-dependent oligomers, both in bryophytes and other plant species.

## Materials and methods

### Plant tissue and growth conditions

For all assays, plants were grown in a Percival growth chamber under constant light conditions at 25˚C. Transgenic plants were made using a *DR5:DsRed* line originally made in the Gransden ecotype and backcrossed to the Reute ecotype of *P. patens* 3 times [17]. Formulations of BCD (AT) media were used as described previously [31]. Transformations of *P. patens* were performed via the PEG-mediated method described previously [32] with the following modifcations: PEG and DNA were added to protoplasts suspended in 3M solution for 10 min prior to heat shock. After heat shock, the cells were allowed to cool to room temperature before approximately 9 mL of 8.5% mannitol was added to dilute the PEG/3M solution. Cells were allowed to recover for 30 to 60 min before plating.

### Mutant screen

The *DR5:DsRed2/NLS-4* strain (*DR5:DsRed/35S:aacC1* inserted into *Pp108B* between *Pp3c20_980* and *Pp3c20_1020*; *2×35S:NLS-GFP-GUS/35S:nptII* inserted randomly) was sterilely homogenized in water in a Waring blender with an autoclaved MC1 mini container, spread onto BCDAT plates containing 100 mg/l gentamicin and 25 mg/l G418 overlain with 90 mm cellophane sheets (AA Packaging) and grown for 1 week. In the course of this work, it was discovered that the gentamicin resistance cassette, *35S:aacC1*, confers resistance to both gentamicin and G418. Plates were then placed in a Stratalinker UV Crosslinker (Stratagene) and, with lids removed, irradiated with doses of 254-nm light. The plates were closed, taped, and stored in the dark for 24 h to block photoreactivation of pyrimidine dimers. For the UV dose-response evaluation, 10 doses ranging from 0 to 500 mJ/cm$^2$ were tested before a dose of 150 mJ/cm$^2$ was chosen for mutagenesis because it caused a mortality rate of approximately 40%. After UV and dark treatments, the cellophanes were transferred to BCD plates with 15 μM NAA (dose-response treatments), 1 μM NAA ("narB" round), or 5 μM NAA ("narC" and "narD" rounds), and 5 μM NAA proved to be the most effective concentration and all mutants were isolated from 12 (narC) and 18 (narD) plates. After 2 to 4 weeks, the plates were screened for mutants that produced green protonemata and/or leafy gametophores. A few cells of 224 candidate mutants were spotted onto new BCD plates containing 5 μM NAA and reevaluated after 2 weeks. This yielded 57 mutants (with 15 additional candidates selected later based on reduced *DR5* expression). The initial 57 mutants were screened for degron mutations in all 3 *AUX/IAA* genes (*IAA1a*/non-annotated, *IAA1b*/*Pp3c8_14720*, and *IAA2*/*Pp3c24_6610*) and loss-of-function mutations in *DIAGEOTROPICA* (*DGT*/ *Pp3c1_32280*) by sequencing PCR products. Eighteen new alleles of these loci were identified: *narC32, dgt-splice* (g to a 7 bp before ATG); *narD138, dgt-S117L*; *narD154, dgt-S106C*; *narC15, iaa1a-P323L*; *narD69, iaa1a-P323L*; *narD75, iaa1a-P323L*; *narD148, iaa1a-G320S* (weak); *narD151, iaa1a-P323L*; *narD153, iaa1a-P323L*; *narC33, iaa1b-P342F*; *narC37, iaa1b-P341L*; *narD68, iaa1b-P342L*; *narD152, iaa1b-P341L*; *narC8, iaa2-P328L*; *narC10, iaa2-P328L*; *narC36, iaa2-P328F*; *narD139, iaa2-P328L*; *narD187, iaa2-P327S*. The remaining mutants were compared after growth on BCDAT, BCD, BCD + 0.5 μM NAA, and BCD + 12.5 μM NAA, and 12 were selected based on growth and *DR5* expression phenotypes.

### Sequencing

Genomic DNA was isolated using the *Quick*-DNA Plant/Seed Miniprep Kit (Zymo), and 7.1 to 8.5 total gigabases of paired-end 100 bp reads (BGI Americas) were generated for each mutant plus the *DR5:DsRed/NLS4* parent strain. Using Bowtie 2 with "end-to-end" and "very-

sensitive" settings (v2.3.4.3; [33]), the reads were aligned to the Gransden reference sequence to which the *DR5:DsRed* transgene plus scaffold_1157 appended with the correctly assembled *IAA1a* gene were added. The SAM files were converted to sorted BAM files using SAMtools and variants were called using BCFtools (v1.9; [34]). The variants were mapped relative to the annotations (CoGe GID 33928 v3; [35]) using SnpEff (v4.3t; [36]).

## Cloning and molecular techniques

For ARF cloning, cDNA was synthesized from RNA extraction via RNeasy plant mini kit (Qiagen) and SuperScript III reverse transcriptase (Thermo Fisher). Coding sequences were amplified and cloned into pENTR/D-TOPO (Thermo Fisher), and subsequently cloned into *pGILDA* and *pB42AD* yeast-2-hybrid vectors (Clonetech) via LR Gateway Reactions (Thermo Fisher). To introduce T653L and Y643* mutations into *ARFd1* entry clone, standard oligomer-driven mutagenesis protocols were used. See S4 Fig for oligomer sequences. For *arfd*$^{dub}$ complementation, *ARFd1* truncations and domain swaps were cloned with GoldenGate cloning. Both *ARFd1* (full length) and *PB1*$^{d1}$ (1,470 to 2,250 bp) sequences were amplified from pENTR-ARFd1 introducing BsaI and unique 4-nucleotide overhangs via PCR. For *ARFd1*$^{b2}$, we generated 3 GoldenGate-ready fragments via PCR off coding sequences cloned into pENTR- 1–778 bp of *ARFd1*, 1,214 to 1,910 bp of *ARFb2*, and 1,470 to 2,250 bp of *ARFd1*. All coding sequence fragments were arranged via GoldenGate with pre-cloned N-terminal mYPet and ZmUbi promoter. The destination plasmid contained a Hsp terminator, 35S:*nptII* resistance cassette, and homology arms to target the construct to the *PIG1* locus. Constructs were transformed into the *arfd*$^{dub}$#3 line with a CRISPR/Cas9 plasmid targeting the *PIG1* locus. Trasformants were selected for with Kanamycin sulfate, and stable lines were screened for expression via mYPet signal.

## Split YFP

To generate split YFP plasmids, we amplified the N-terminal and C-terminal split YFP cassette described in [37]. We cloned each fragment into *pMiniT* using the PCR Cloning Kit (New England BioLabs), following the manufacturer's instructions. The resulting plasmids (*pMnVN-Gate* and *pMnVC-Gate*) made N-terminal fusions of either the N or C terminal half of mVenus to our protein-of-interest without the excess sequences needed for *Agrobacterium* transformation. Given that each plasmid is Gateway (Thermo Fisher) compatible, we were able to use LR reactions with the same entry clones used in Y2H described above. We then transformed 15 µg of each plasmid into *P. patens* protoplasts. These protoplasts were set to incubate in the light for approximately 60 h before imaging via confocal microscopy.

## Chlorophyll extraction

Methanol chlorophyll extractions were performed at room temperature overnight in 2 mL Eppendorf tubes on whole tissue pressed lightly between paper towels to remove excess water. Absorbances were measured on a SmartSpec Plus spectrophotometer. Chlorophyll content was calculated via the following equation, as described previously [38]:

$$\mu g_{Chl}/g_{tissue} = ((-8.9062^*(A_{652nm} - A_{750nm})) + (16.5169^*(A_{665nm} - A_{750nm})))/g_{tissue}.$$

## Image acquisition, analysis, and preparation

Images of spot inoculations were taken with a Nikon SMZ1500 microscope and a DS-Ri1 camera. For the colony morphology experiments, pictures were acquired via flatbed scanner. Shape parameters were measured in ImageJ based on a binary mask made from color pictures of plants. For regenerated protoplast assays, images were acquired with BZ-X810 all-in-one microscope (Keyence) using CFP and TRITC filter cubes. Whole plant area was measured in ImageJ using a mask generated from the signal from the calcofluor cell wall stain (Fluorescent Brightener 28, Sigma). Cell length, cross-wall angle, and branch positioning were manually measured in ImageJ. Student's $T$ tests were performed in Excel (Microsoft), and the Kolmogorov–Smirnov, ANOVA, and TukeyHSD test was done with R. Confocal microscopy was performed on a Zeiss LSM 880, with image processing and quantification performed in ImageJ.

Dot plots were generated with an internet application [39]. Bar plots and histograms were made in Excel (Microsoft). Figures were prepared with Affinity Designer (Serif Europe).

## qPCR

cDNA generated from 3 biological replicates for each condition, with 2 technical replicates each, were run on a CFX Connect Real Time System thermocycler/plate reader (BioRad). A *P. patens ADENINE PHOSPHORIBOSYLTRANSFERASE* (*APT*) gene (*Pp3c8_16590V3.1*) was used as a reference housekeeping gene as described [40].

## Supporting information

**S1 Fig. Schematic of *arfd* mutations.** (A) Models of *PpARFd1* and *PpARFd2*, including the targets of CRISPR/Cas9-mediated knock out. (B) Schematic of *PpIAA2* and *PpAFB* mutations. (A) A single nucleotide mutation (red text) in the degron (DII) motif in *PpIAA2* results in a stabilized auxin repressor. (C) Alleles recovered when generating *afb1,2,3,4* line. Green text indicates endogenous start codon. Start codon was knocked out in *PpAFB3*. The 2 nt deletion in *PpAFB4* generates a frameshift mutation, resulting in a premature stop codon. The *afb4* peptide is 21 amino acids long.
(PDF)

**S2 Fig. *arfd* phenotypes are minimized on BCD medium.** (A) Images of 21-day-old spot inocula on BCD medium. (B) Normalized chlorophyll content of 21-day-old colonies grown on BCD supplemented with the indicated concentration of NAA. (C) Gametophore counts of 21-day-old colonies grown on BCD supplemented with the indicated concentration of NAA. For both B and C, $n \geq 18$ across 3 replicates. Error bars are standard deviation. (D) Morphology of 21-day-old colonies of *arfd* mutants are similar on low-nitrogen BCD media. On BCDAT, however, the *arfd^{dub}* are significantly more circular. Letters indicate statistical groups as determined by a TukeyHSD post hoc test of an ANOVA. (E) RSL6, an auxin-induced gene, is significantly up-regulated in tissue grown on BCD for 3 days. Expression level normalized to gene expression on tissue kept on BCDAT for 3 days. IAA2 is nominally down-regulated in repsonce to BCD treatment, while ARFd1 is not differentially expressed. Bar graphs indicate mean of 3 biological replicates, error bars are standard error of the mean. ** = $p < 0.005$. The underlying data for panels B, C, D, and E are in S1 Data. Representitive images for colony morphology image analalysis are in S2 Data.
(PDF)

**S3 Fig. Schematic of mutagenesis via CRISPR/Cas9 and mutating oligonucleotide.** (A) Gene diagram of *pARFd1:PpARFd1-mYPet* locus with the PB1 domain in blue. Indicated location of sgRNAs (red text) used to create T653L and Y643* mutations. Scale bar is 1,000 nts. (B)

Sequences of WT, *narD122*, mutating oligomer, and recovered *arfd1*$^{T653L}$ or *arfd1*$^{Y643*}$ mutants. Magenta text highlights codon change, with green highlighting silent mutations intended to introduce restriction enzyme sites. Red/pale red text indicates the protospacer-adjacent motif (PAM), while orange text highlights silent mutations introduced to abolish sgRNA binding. For *arfd1*$^{T653L}$, only the PAM sequence needed to be changed. For *arfd1*$^{Y643*}$, there were no available silent mutations for the lysine residue that would not also disrupt the intron-exon slice site. (C) Micrographs of ARFd1-mYPet and 2 independent lines of arfd1$^{T653L}$-mYPet. Magenta is chloroplast autoflouresence, yellow is mYPet signal. Often, both ARFd1 and arfd1$^{T653L}$ formed single subnuclear focus (white arrow head) and scale bar is 5 μm. (B) mYPet signal, normalized to ARFd1-mYPet. Signal of arfd1$^{T653L}$-mYPet ($n \geq 66$ cells across 2 replicates) is roughly double that of ARFd1 ($n = 170$ cells across 2 replicates). * = $p \leq 0.001$. The underlying data for panel D are in S1 Data. Representitive images describing nuclear signal image analaysis are in S2 Data.
(PDF)

**S4 Fig. Strong auxin signaling mutants exhibit ammonium-dependent caulonemal differentiation.** (A) Representative micrographs of 7-day-old plants regenerated from single protoplasts. WT, *iaa2-mDII*, and *afb1,2,3,4* each exhibit changes in morphology in response to ammonium in the medium. Scale bar is 500 μm. (B) Cross-wall angles of protonemata from WT ($n = 660$), *iaa2-mDII* ($n = 631$), and *afb1,2,3,4* ($n = 587$) plants across 3 biological replicates grown on BCD medium. Protonemata from both mutant lines have a smaller average cross-wall angle compared to wild type (Student's *T* test, $p < 0.001$. Error bars are standard deviation.). (C) Protonemata form both lines are enriched in chloronemata (Kolmogorov–Smirnov test, $p < 0.001$). (D) Cross-wall angles of protonemata from WT ($n = 583$), *iaa2-mDII* ($n = 583$), and *afb1,2,3,4* ($n = 560$) plants across 3 biological replicates grown on BCDAT medium (Student's *T* test, $p < 0.001$. Error bars are standard deviation.). (E) Protonemata form both lines are enriched in chloronemata (Kolmogorov–Smirnov test, $p < 0.001$). The underlying data for panels B, C, D, and E are in S1 Data. Representitive images demonstrating cross-wall image analaysis are in S2 Data.
(PDF)

**S5 Fig. Protonemal branching is an auxin-driven developmental processes.** (A) Complete table of values for *arfd* mutants grown on (+) or off (-) 5 μM IAA for 3 days. (B) Representative micrographs of protonemal filaments of 7-day-old regenerated protoplasts. White arrowhead indicates filament branch. Scale bar 50 μm. (C) Protonemal branching occurrence by cell position. Plants were scored after growing for 3 days on BCD, having regenerated from single protoplasts on PRMB for 4 days. There is a significant delay in the stabilized *iaa2-mDII* ($n = 228$ cells) line as well as a line lacking any F-box auxin co-receptor, *Δafb1,2,3,4* ($n = 144$ cells) compared to wild type ($n = 329$ cells) (K.S. test, $p < 0.001$, 4 replicates for each line). (D) Complete table of values of data presented in (C). The underlying data for panels C and D are in S1 Data.
(PDF)

**S1 Data. Raw data that contributed to Figs 2B–2E, 3B and 3C, 4B–4D, 5H, 6B, 8B and 8C, S2B–S2E, S3D, S4B–S4E, S5C and S5D Figs.**
(XLSX)

**S2 Data. Representative annotated micrographs for the image anaylsis done in this paper.** (A) Representive cross-wall measurements. Scale bar is 5 μm. (B) Representive cross-wall measurements and Integrated Density (ID) quantification of ARFd1-mYPet signal in the nucleus (white outline). Scale bar is 5 μm. (C) Representitive example of imaging processing for colony morphology quantificiation. (D) Representitive micrographs of nuclear ARFd1-mYPet and

ARFd1$^{T653L}$-mYPet signal quantification. White outline is ROI for nucleus. Red outline is ROI for background subtraction. Only 1 background ROI was measured for each micrograph containing multiple nuclei. Scale bar is 5 μm.
(TIFF)

## Author Contributions

**Conceptualization:** Mark Estelle.

**Data curation:** Carlisle Bascom, Jr.

**Formal analysis:** Carlisle Bascom, Jr.

**Funding acquisition:** Mark Estelle.

**Investigation:** Carlisle Bascom, Jr., Michael J. Prigge.

**Methodology:** Carlisle Bascom, Jr., Michael J. Prigge.

**Resources:** Whitnie Szutu, Alexis Bantle, Sophie Irmak, Daniella Tu.

**Supervision:** Carlisle Bascom, Jr., Michael J. Prigge, Mark Estelle.

**Visualization:** Carlisle Bascom, Jr.

**Writing – original draft:** Carlisle Bascom, Jr., Michael J. Prigge.

**Writing – review & editing:** Mark Estelle.

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
