## [Editor Report · Decision Letter 0]

17 Nov 2022

Dear Mark, 

Thank you for submitting your manuscript entitled "Clade-D Auxin Response Factors are Major Regulators of Auxin Signaling in Physcomitrium patens." for consideration as a Research Article by PLOS Biology.

Your manuscript has now been evaluated by the PLOS Biology editorial staff as well as by an academic editor with relevant expertise and I am writing to let you know that we would like to send your submission out for external peer review.

Once your full submission is complete, your paper will undergo a series of checks in preparation for peer review. After your manuscript has passed the checks it will be sent out for review. To provide the metadata for your submission, please Login to Editorial Manager (https://www.editorialmanager.com/pbiology) within two working days, i.e. by Nov 21 2022 11:59PM.

Kind regards,

Ines

--

Ines Alvarez-Garcia, PhD

Senior Editor

PLOS Biology

---

## [Decision Letter · Decision Letter 1]

22 Dec 2022

Dear Mark,

Thank you for your patience while your manuscript entitled "Clade-D Auxin Response Factors are Major Regulators of Auxin Signaling in Physcomitrium patens" was peer-reviewed at PLOS Biology. It has now been evaluated by the PLOS Biology editors, an Academic Editor with relevant expertise, and by three independent reviewers. 

The reviews are attached below. As you will see, the reviewers find the conclusions interesting and worth pursuing for publication. Nevertheless, they also raise several points that need to be addressed and suggest additional analyses to confirm some of the findings. After consulting with the Academic Editor, we would like to invite you to revise the work to thoroughly address the reviewers' comments.

Given the extent of revision needed, we cannot make a decision about publication until we have seen the revised manuscript and your response to the reviewers' comments. Your revised manuscript is likely to be sent for further evaluation by all or a subset of the reviewers.

**IMPORTANT - SUBMITTING YOUR REVISION**

3. Resubmission Checklist

a) *PLOS Data Policy*

b) *Published Peer Review*

Sincerely,

Ines

--

Ines Alvarez-Garcia, PhD

Senior Editor

PLOS Biology

Reviewers’ comments

Rev. 1:

The manuscript by Bascom Jr. et al. presents a functional analysis of the D-class ARF proteins in Physcomitrium patens. It combines a genetic approach following a genetic screen for NAA-resistant (NAR) phenotypes, with phenotyping, expression and protein interaction analysis. It is neat and straightforward.

The authors identified in the screen three alleles in ARFd1 and constructed the other alleles by CRISPR, and the double mutant d1 d2 also by CRISPR.

Here I was confused with the labeling of the genotype in Fig S1 and Figs 2, 3, 4 (and others). In panel A of S1, are presented the gene deletion in arfd2-8, d2-8 d1-1 and d2-8 d1-3. But in panel B, the images are from d1 d2-1 and d1 d2-3. Can the authors checking the accuracy of the labeling in Fig S1 and 2? Maybe a table can be proposed to detail the genotypes used in the study?

The authors analysed the effects of the mutations in the PB1 domain, the effects of NAA applications and high-nitrogen. They showed that there is an impact of the mutations of the timing of development of the branches.

To further understand how D1 can act on activate transcription without having a DNA binding domain, the authors investigated its interaction with A-ARF and IAA1, by yeast two-hybrid and BCIF, proposing a model that D-ARF act as an enhancer of the activity of A-ARF on the promoters of auxin-responsive genes.

The authors indicated that D1 also interacts with A4, while only A8 is shown in Fig 7. I am also curious if the authors looked into the expression pattern (tissue and cell level) of the arfd1 T653L - mYpet. Is the mYPet protein expressed? Is there any differences with the WT D1-mYPet?

These are only changes amendments. The manuscript is in state to be published when this is clarified.

Rev. 2:

In this manuscript, Bascom and colleagues address the function of Clade-D ARFs, a class of ARFs missing an obvious DNA-binding domain which is found only in bryophytes and lycophytes among extant land plants. To this end, they subject the Physcomitrium patens genes ARFd1 and ARFd2 to a functional characterization.

The manuscript is well written and to-the-point. Despite a relatively limited set of experiments and a compact format the authors show that clade-D ARFs act as regulators of physiologically significant auxin signaling in moss and also present a plausible model for their mode of action. Taken together, this is a major contribution of considerable news value with potential to attract the interest of colleagues studying plant development, land plant evolution and auxin function.

The authors should consider the following comments and suggestions to further strengthen the manuscript (ordered according to structure of manuscript rather than priority):

1. I have no questions, comments or suggestions relating to the title, abstract, or introduction.

2. Relating to Fig 1:

a. The authors could consider to remove narC18 and narD157 from the figure since the more severe phenotypes of these lines are not primarily related to mutations in ARFd genes.

b. The phenotypic deviation in presence of NAA in narD122, nard122/Δarfd1, and Gd/NLS4Δarfd1 (marked by arrow heads) cannot really be seen. The authors should include inserts of gametophores at higher magnification.

c. Why does the DR5:DsRed2 output appear much stronger in Gd/NLS4Δarfd1 than narD122 and nard122/Δarfd1? Please comment.

d. I suggest that also DR5:DsRed2 output in the absence of NAA is shown. Please add or explain why it is not needed.

3. Relating to Fig S1:

a. The results section states that 'we used CRISPR/Cas9 to delete ARFd2 in the WT and Δarfd1 background', but when I look at Fig S1A I get the impression that ARFd1 was targeted in a pre-existing Δarfd2-8 background. Please change or clarify.

b. I would have liked to see the phenotypic characterization presented in Fig S1B-D and Fig 2 repeated for more independent single and double mutant lines. If such data exist, please include it. If not, I still find it acceptable in this particular case since additional and partly supporting phenotyping data is presented in other figures for other mutations in the same genes.

4. Relating to Fig 3:

a. In the results section discussing Fig 3b it is stated that 'arfd1T653L-mYpet lines exhibit an auxin sensitivity that is intermediate between the control and Δarfd1,d2 lines'. This is not very obvious from the data. Please provide statistical support for this claim.

b. Fig 3C indicates lower gametophore counts in arfd1Y543*-mYPet Δarfd2 lines than in the Δarfd1,d2 lines. Is there an explanation for this?

5. Relating to Fig 4:

a. The finding that ARFds are required for caulonemal development only in nutrient-rich conditions is interesting. Can it imply that ARFd serves as a modifier of auxin responsiveness in response to nutrient availability? Maybe it could be worth to check if the expression of ARFd (and possibly auxin regulated genes likely to act downstream of ARFd) is regulated by the nitrogen source?

6. Relating to Fig 5 and the corresponding results section:

a. The data supporting that auxin affects the timing of branch initiation in Physcomitrium appears very solid. Still, after a quick look at Johri and Desai (1973) my impression is that their claim is that the proportion of caulonema-like branches is increased in response to auxin in Funaria, rather than the number of branches or the timing of branching. I may be wrong, but the author may want to comment on this.

b. The data presented appears to clearly support that branch initiation is delayed in the Δarfd1,d2 lines both in absence and presence of external auxin, but it is not as obvious if the ARFd genes are important for mediating the auxin-induced change in timing of branch initiation. The authors may want to provide statistical support for such a claim.

7. Relating to Fig 6:

a. The strong ARFd expression in protonemal cells at the base of gametophore initials is intriguing. An obvious question that comes to mind is if ARFd expression maxima in certain protonemal cells marks where branches differentiating into buds/gametophores will emerge. It is not a trivial experiment, but if the authors could provide data supporting such a function by some kind of time-lapse study it would be a major contribution.

8. Relating to Figure 7:

a. In the results it is stated that ARFd1 interacts with both ARFa4 and ARFa8, but I cannot find data supporting the former claim.

b. In the results it is stated that ARFd1ΔDIV retains the ability to interact with ARFa8 but not with ARFb1, but I cannot find data supporting the latter claim.

9. I find the discussion clear, relevant and to-the-point. I don't have any questions, comments or suggestions relating to the Material and Methods section.

Rev. 3: Dolf Weijers – note that this reviewer has signed his review

Bascom Jr and colleagues describe a genetic analysis of the Physcomitrium patens D-class ARFs. This is a mysterious subclade of the ARF transcription factor family (also referred to as non-canonical ARFs; ncARFs) that exist in bryophytes and lycophytes, but were lost in ferns and seed plants. A prior study in Marchantia showed that a D-class ARF mutant has reduced auxin response, but no detailed analysis was perfoemd, and no mechanism has been established. The authors of the present manuscript identified a D-class ARF mutant in a screen for auxin-resistant Physcomitrium mutants, and explore its role in auxin response. Overall, this reports important findings that are of importance for understanding the principles underlying auxin response. However, there are a number of points that require further exploration, clarification or elaboration.

1. The authors present a role of ARFds during P. patens development. However, unlike the authors suggest in the abstract, it does not seem to be "major role", given that this occurs only under specific nutritional conditions.

2. While the "ARFd" nomenclature was coined by the authors (Lavy et al. 2016), the term D-class has not been widely used. Instead, ncARF (non-canonical ARF) has been used in the literature. I encourage the authors to explicitly state that this refers to the same proteins and propose the future used of only one name (ARFd).

3. The interest in D-class mutations is clear, but the reader can be left wondering about the other mutations found in the mutagenized lines, as those in A and B-class ARFs, PINc or RSL7. I suggest to either leave out the information on the (putative) identity of these other mutants, or provide information on the molecular lesions, especially given the difference in the visible phenotypes in Fig.1 for narC18 and narD157.

4. Similarly, while the authors perform CRISPR against ARFd1 in the narD122 line, this line alone is not proven to be complemented with a ARFd1 WT copy, so other mutations, that most likely have occurred due to the technique used, may be important for auxin sensitivity (maybe epistatic to ARFd1?). What other mutations were found in narD122?

5. The authors make single and double arfd mutants, but quickly jump to chlorophyll content analyses. Are these mutants similar or different to WT? how do they look in BCD and BCDAT with or without different concentrations of auxins? Some of this data is in supplemental material but is never described or discussed.

6. Image quality in Fig. 1 and 2 is poor. Please provide better pictures, maybe higher magnification would convey the phenotypes better. This would help the reader in following authors´ interpretation for auxin resistance.

7. Authors only show DR5:dsRED expression in response to NAA. In order to properly compare DR5:dsRED expression it would be important to see DR5:dsRED activity on medium without auxin, and ideally also with IAA (rather than NAA).

8. In some cases, the DR5 line shows a somewhat different behaviour in auxin response compared to a WT. For example, the DR5 strain shows a much smaller reduction in chlorophyll content (22% vs 36%) compared to the WT strain. The authors describe this as "similar", but this difference is striking and not trivial. Also, together with other findings, it seems that this background has inherent auxin insensitivity. Is this something common in Reute/Grandsen cross progenies? Why not making mutants in a non-DR5 background since the responsiveness seems to be dampened here?

9. In conjunction with the comment above, I have some issues with using chlorophyll content as a way of quantifying auxin response. The authors mention later that arfd1,2 mutations lead to delays in gametophore formation and protonema differentiation on BCDAT. Doesn't this fact introduce a bias in the chlorophyll analysis? Ater 2 weeks, the composition of tissues in the mutants is very different form wild-type (i.e. WT should have more gametophores, whereas the arfd1,2 mutant should have relatively more chloronema). These different tissues would contribute to the overall chlorophyll content per gram of tissue in a unique way. Are these stages comparable?

10. It is unclear why the authors did not include the original arfd/nard or arfd2 single mutants in the gene expression analyses (Fig. 2C). These single mutants are disregarded also from Fig. 3 on. Also, why using such a long IAA treatment (4.5 hours) to measure immediate auxin responsive gene activation? The authors indicate that ARFd1 has a larger contribution to the auxin response than ARFd2. But this is entirely bound to the first chlorophyll content analysis and single mutants are not checked for any other analysis. Is there any specific reason?

11. The results suggest that ARFd proteins are involved in altering growth patterns in response to changing nutrient conditions. To provide mechanistic understanding, it would be important to know if (and how) ARFd expression and protein levels are modulated by the nutrient contrasts used here.

12. ARFd1-mYPet expression seems to be very similar between chloronema and caulonema tissues. Since ARFd1 appears to also have an impact on timing of branch formation, it would be interesting to know if ARFd1 expression levels are different in the apical and subapical cells, and how this correlates with branch formation. Authors also claim that YPet is not detected in mature gametophores, but the picture is not clearly showing that, especially at this magnification, since it could still be expression at the basal parts of the mature gametophore.

13. An open question is the molecular mechanism by which ARFd acts. One option is that it supports transcriptional activation by A-class ARFs, another is that is acts as an Aux/IAA scavenger (as proposed by Mutte et al, 2018). The authors clearly favor the former throughout the text, and do not comment on the possibility of the latter at all. I would encourage the authors to use e.g. transient gene expression assays in protoplasts to test these two hypotheses. Otherwise, the existence of a second hypothesis should be mentioned in the discussion, and the claim of co-activation activity toned down.

14. In page 20, the authors write: "This is the first evidence in any system that ARF oligomerization is important for activity.". This statement is not entirely correct. Kato et al 2020 showed that the MpARF1 PB1 domain is essential for function, and that it can be replaced by a heterologous oligomerization domain. I would suggest removing this statement.

15. At the end of page 21, the authors write: "…most models suggest that ARF-ARF interaction occurs primarily via the dimerization domain adjacent to the DBD, and only when the proteins are bound to DNA (Boer et al., 2014)". This is also not entirely correct, ARFs dimerize through their DBD also in the absence of DNA. Moreover, a recent preprint showed that the DBD and PB1 act cooperatively in promoting dimeric DNA binding, indicating that both domains contribute to dimer strength.

16. The text needs some refinement to improve clarity:

o In the introduction, the authors refer to the DNA binding sequence of ARFs. While the shown sequence is indeed an ARF binding motif, this not the only one. This may lead to confusion for a general audience. I suggest to reword to auxin responsive elements, but given these specific piece of data adds nothing to the rest of the work, the sentence could be deleted.

o Authors introduce only A- and B-class at the beginning but then refer to C-class. This can also be wrongly interpreted.

o Mutant nomenclature is not clear. While the single alleles are named properly in Fig. S1 (as with Δarfd2-8), the double mutants obtained are also dashed in the text and figures (Δarfd1,d2-1). This dash is misleading, especially since it refers to the ARFd1 allele.

o Authors indicate in page 12 the use of NAA for the screening, but later in page 13 explain why they use NAA instead of IAA. This should be explained at the very beginning.

o On page 14 the authors mention Figure 3C, D but Fig. 3D does not exist (it refers to Fig. 3B, C).

o On page 13 the authors say that "Previous reports that auxin treatment results in down regulation... photosynthesis", please provide a reference.

o On page 14 the sentence "To gain insight...narD122" would read easier if the authors used the term "mutation" instead of "mutant".

o In Figure 4 it would be much easier to interpret results if the type of medium used to grow plants is mentioned IN the figure (BCD resp. BCDAT) instead of in the figure description.

o Figure 5 D, G and J contain the same data replicated. While I guess this done for clarity, please explain or indicate this in the captions for the sake of transparency.

o Unmerge K-S tests in Fig. 5G and 5H for each of the mutant lines. Each line seems to behave differently (in mock conditions).

o In general, captions are quite uninformative and lack data for proper figure interpretation such as age of plants, time of treatments, etc. Statistical tests, number of samples, and replicates should be shown as well.

---

## [Decision Letter · Decision Letter 2]

25 Apr 2023

Dear Mark,

Thank you for your patience while we considered your revised manuscript entitled "Clade-D Auxin Response Factors Regulate Auxin Signaling and Development in Physcomitrium patens" for publication as a Research Article at PLOS Biology. This revised version of your manuscript has been evaluated by the PLOS Biology editors, the Academic Editor and two of the original reviewers.

Based on the reviews, we are likely to accept this manuscript for publication, provided you satisfactorily address the data and other policy-related requests stated below.

In addition, we would like you to consider a suggestion to improve the title:

"Clade-D auxin response factors regulate auxin signaling and development in the moss Physcomitrium patens"

We expect to receive your revised manuscript within two weeks. 

*Published Peer Review History*

*Press*

Sincerely,

Ines

--

Ines Alvarez-Garcia, PhD

Senior Editor

PLOS Biology

Fig. 2B-E; Fig. 3B, C; Fig. 4B-E; Fig. 5H; Fig. 6B; Fig. 8B, C; Fig. S2B-E; Fig. S3D; Fig. S4B-E and Fig. S5C

Reviewers' comments

Rev. 2:

All my requests and suggestions on the original submission have been addressed in an acceptable way.

Rev. 3: Dolf Weijers

The authors have done an excellent job in responding to my comments and revising the manuscript. I look forward to seeing this paper in print.

---

## [Editor Report · Decision Letter 3]

12 May 2023

Dear Mark,

Thank you for the submission of your revised Research Article entitled "Clade-D Auxin Response Factors Regulate Auxin Signaling and Development in the moss Physcomitrium patens" for publication in PLOS Biology. On behalf of my colleagues and the Academic Editor, Xinnian Dong, I am delighted to say that we can in principle accept your manuscript for publication, provided you address any remaining formatting and reporting issues. These will be detailed in an email you should receive within 2-3 business days from our colleagues in the journal operations team; no action is required from you until then. Please note that we will not be able to formally accept your manuscript and schedule it for publication until you have completed any requested changes.

PRESS

Best wishes, 

Ines

--

Ines Alvarez-Garcia, PhD

Senior Editor

PLOS Biology
